A new glassfrog (Centrolenidae) from the Chocó-Andean Río Manduriacu Reserve, Ecuador, endangered by mining

http://orcid.org/0000-0003-0098-978X Guayasamin Juan M. 1 2 jmguayasamin@usfq.edu.ec
http://orcid.org/0000-0002-6132-2738 Cisneros-Heredia Diego F. 3 4
Vieira José 1 5
Kohn Sebastián 6 7
Gavilanes Gabriela 1
Lynch Ryan L. 8
Hamilton Paul S. 9
Maynard Ross J. 9
1 Laboratorio de Biología Evolutiva, Colegio de Ciencias Biológicas y Ambientales COCIBA, Instituto BIÓSFERA-USFQ, Campus Cumbayá, Universidad San Francisco de Quito USFQ , Quito , Ecuador
2 Centro de Investigación de la Biodiversidad y Cambio Climático, Ingeniería en Biodiversidad y Recursos Genéticos, Facultad de Ciencias del Medio Ambiente, Universidad Tecnológica Indoamérica , Quito , Ecuador
3 Colegio de Ciencias Biológicas y Ambientales COCIBA, Instituto de Zoología Terrestre y Museo de Zoología, Instituto de Geografía, Universidad San Francisco de Quito USFQ , Cumbayá , Ecuador
4 División de Herpetología, Instituto Nacional de Biodiversidad INABIO , Quito , Ecuador
5 Tropical Herping , Quito , Ecuador
6 Fundación Cóndor Andino , Quito , Ecuador
7 Fundación EcoMinga , Baños , Ecuador
8 Third Millennium Alliance , Fremont, CA , USA
9 The Biodiversity Group , Tucson, AZ , USA
Gillespie Joseph
Electronic publication date: 2019 Feb 26
Publication date: 2019
Volume: 7
Electronic Location ID: e6400
Received 2018 Oct 2; Accepted 2019 Jan 6
Copyright: © 2019 Guayasamin et al.
Copyright year: 2019
Copyright holder: Guayasamin et al.
License: This is an open access article distributed under the terms of the Creative Commons Attribution License, which permits unrestricted use, distribution, reproduction and adaptation in any medium and for any purpose provided that it is properly attributed. For attribution, the original author(s), title, publication source (PeerJ) and either DOI or URL of the article must be cited.
License URL: https://creativecommons.org/licenses/by/4.0/

Keywords: Taxonomy, Conservation, Glassfrog, Endangered, Mining, Amphibia

Funding: SENESCYT Programa Becas de Excelencia USFQ Collaboration Grants, Chancellor Grants, and Research Funds of projects 34, 41, 48, 1057 SENESCYT project PIC-0000470, Programa Nacional de Financiamiento de Investigación INÉDITA: Respuestas a la crisis de biodiversidad: la descripción de especies como herramientas de conservación Universidad San Francisco de Quito Collaboration Grants 5521, 5467, 5447, 11164, Fondos COCIBA and Fondos Semillas Biosfera This study was supported by SENESCYT (Programa Becas de Excelencia) and USFQ (Collaboration Grants, Chancellor Grants, and Research Funds of projects 34, 41, 48, 1057) to Diego F. Cisneros-Heredia, and by SENESCYT (project PIC-0000470, Programa Nacional de Financiamiento de Investigación INÉDITA: Respuestas a la crisis de biodiversidad: la descripción de especies como herramientas de conservación), and Universidad San Francisco de Quito (Collaboration Grants 5521, 5467, 5447, 11164, Fondos COCIBA and Fondos Semillas Biosfera) granted to Juan M. Guayasamin. The funders had no role in study design, data collection and analysis, decision to publish, or preparation of the manuscript.

==============================
We describe a new glassfrog from Río Manduriacu Reserve, Imbabura Province, on the Pacific slopes of the Ecuadorian Andes. The new species can be distinguished from most other glassfrogs by having numerous yellow spots on the dorsum and lacking membranes among fingers. Both morphological and molecular data support the placement of the species in the genus Nymphargus. We present a new mitochondrial phylogeny of Nymphargus and discuss the speciation patterns of this genus; most importantly, recent speciation events seem to result from the effect of the linearity of the Andes. Finally, although the new species occurs within a private reserve, it is seriously endangered by mining activities; thus, following IUCN criteria, we consider the new species as Critically Endangered.

Introduction

Glassfrogs represent one of the most charismatic Neotropical radiations (see Guayasamin et al., 2009; Hutter, Guayasamin & Wiens, 2013; Castroviejo-Fisher et al., 2014). This clade contains more than 150 species, with an amazing variety of morphology (Cisneros-Heredia & McDiarmid, 2007; Guayasamin et al., 2009), behavior (Delia, Bravo-Valencia & Warkentin, 2017; Delia et al., 2018), and unresolved taxonomic problems.

Within the tropical Andes, the most diverse glassfrog genus is Nymphargus (Cisneros-Heredia & McDiarmid, 2007, as modified by Guayasamin et al., 2009), a taxon previously defined as the Cochranella ocellata Group (Ruiz-Carranza & Lynch, 1991, 1995). Nymphargus is a monophyletic taxon containing 36 species (Frost, 2018). This Andean genus is primarily composed of species with restricted distributions and, therefore, it is not unusual to find new taxa in previously unexplored areas (e.g., N. caucanus Rada, Ospina-Sarria & Guayasamin, 2017; N. sucre Guayasamin, 2013; N. buenaventura Cisneros-Heredia & Yánez-Muñoz, 2007).

Recent fieldwork at Río Manduriacu Reserve (RMR, hereafter), Imbabura Province, Ecuador, has revealed a striking new Nymphargus that we describe below. The new species, and two other critically endangered anurans, Nymphargus balionotus (Duellman, 1981) and Rhaebo olallai (Hoogmoed, 1985) are threatened by illegal mining activities within the reserve (see Discussion; Lynch et al., 2014).

Although our work presents several novel components (i.e., new phylogeny of Nymphargus, discussion of biogeographic patterns), we would like to stress the discussion on the conservation problems that the Chocó-Andean forests of Ecuador are currently facing. Ecuador is a megadiverse country (Mittermeier et al., 1999), part of the most diverse hotspot on Earth (the Tropical Andes; Myers et al., 2000), and the country with the highest amphibian species richness per unit of area in the world (600 species in an area of 256,423 km2). However, ecosystems are under heavy pressure from agriculture, wood extraction, oil palm plantation, and, most recently, mining (Roy et al., 2018; Lessmann et al., 2016). Our study, thus, is an attempt to draw attention from the government, NGOs, local communities, scientists, and the general public toward the conservations of the few Chocó-Andean forests still remaining in Ecuador. We also think that taxonomists should play a more active role in conservation biology, mainly because the results of our work (i.e., new species with limited distributions) are powerful tools to justify habitat conservation, specially through partnerships with environmental NGOs.

Materials and Methods

Ethics statement. Research was conducted under permits N°MAE-DNB-CM-2015-2017, 018-2017-IC-FAU-DNB/MAE, 019-2018-IC-FAU-DNB/MAE, issued by the Ministerio del Ambiente del Ecuador. The study was carried out in accordance with the guidelines for use of live amphibians and reptiles in field research (Beaupre et al., 2004), compiled by the American Society of Ichthyologists and Herpetologists, the Herpetologists’ League and the Society for the Study of Amphibians and Reptiles.

Taxonomy and species concept. Glassfrog generic and family names follow the taxonomy proposed by Guayasamin et al. (2009). For recognizing species, we adhere to the General Species Concept (De Queiroz, 2005, 2007). Under this concept, the only necessary property for an entity to be a recognized as a species is that it corresponds to a temporal segment of a metapopulation lineage evolving separately from other lineages (De Queiroz, 2005, 2007). Independent evolution generates traits that can be used to diagnose the species, such as morphology, monophyly, vocalizations, among others.

The electronic version of this article in portable document format will represent a published work according to the international commission on zoological nomenclature (ICZN), and hence the new names contained in the electronic version are effectively published under that Code from the electronic edition alone. This published work, and the nomenclatural acts it contains, have been registered in ZooBank, the online registration system for the ICZN. The ZooBank life science identifiers (LSIDs) can be resolved and the associated information viewed through any standard web browser by appending the LSID to the prefix http://zoobank.org/. The LSID for this publication is: urn:lsid:zoobank.org:pub:E5C0E7E4-9C69-4830-A514-AD1F4B80311C. The online version of this work is archived and available from the following digital repositories: PeerJ, PubMed Central and CLOCKSS.

Morphological data. Morphological characterization follows Cisneros-Heredia & McDiarmid (2007). Webbing nomenclature follows Savage & Heyer (1967), as modified by Guayasamin et al. (2006). We examined alcohol-preserved specimens mostly from the collection at the Instituto de Ciencias Naturales of the Universidad Nacional de Colombia (ICN), Museo de Zoología of the Universidad Tecnológica Indoamérica (MZUTI), Natural History Museum and Biodiversity Research Center of the University of Kansas (KU), and Museo de Zoología of the Universidad San Francisco de Quito (ZSFQ); all examined specimens are listed below. Morphological measurements were taken with Mitutoyo® digital caliper to the nearest 0.1 mm, as described by Guayasamin & Bonaccorso (2004) and Cisneros-Heredia & McDiarmid (2007) except when noted, and are as follow: (1) snout–vent length (SVL); (2) tibia length; (3) foot length; (4) head length; (5) head width; (6) interorbital distance; (7) upper eyelid width; (8) internarial distance; (10) eye diameter; (11) tympanum diameter; (12) radioulna length; (13) hand length; (14) Finger I length; (15) Finger II length = distance from outer margin of palmar tubercle to tip of Finger II; (16) width of disc of Finger III. Sexual maturity was determined by the presence of vocal slits in males and by the presence of eggs or convoluted oviducts in females. Color patterns are described based on photographs of live specimens taken in the field. The adjective “enamelled” is used to describe the shiny white coloration produced by accumulation of iridophores (Lynch & Duellman, 1973; Cisneros-Heredia & McDiarmid, 2007). Examined material is listed in Appendix 1.

Study site. The RMR (0.31°N, 78.85°W, 1,200–2,000 m; Fig. 1) is located at the juncture of the Chocó and Tropical Andes bioregions, near the following Important Bird Areas and Key Biodiversity Areas: Reserva Ecológica Cotacachi Cayapas (EC037), Intag-Toisán (EC038), Bosque Protector Los Cedros (EC039), and Mashpi-Pachijal (EC108; Freile & Santander, 2005).

Figure 1 Map of Ecuador showing the location of Río Manduriacu Reserve, the type locality of Nymphargus manduriacu sp. nov.

Bioacoustics. Sound recordings were made with an Olympus LS-10 Linear PCM Field Recorder and a Sennheiser K6–ME 66 unidirectional microphone. The calls were recorded in WAV format with a sampling rate of 44.1 kHz/second with 16 bits/sample. All calls are stored at the Laboratorio de Biología Evolutiva at Universidad San Francisco de Quito (LBE). Measurements of acoustic variables were obtained as described in Hutter et al. (2013). A call is defined as the collection of acoustic signals emitted in sequence and produced in a single exhalation of air. A note is a temporally distinct segment within a call; notes are separated by a silent interval. Pulsed notes are those having one or more clear amplitude peaks while tonal notes have relatively constant amplitude throughout the call. A call series is defined as a sequence of calls that are separated by a consistent time interval of background noise between calls (see Köhler et al., 2017).

Fieldwork. Sampling at RMR was conducted during the following dates: November 7–8, 2012 (RL, SK), May 13–15, 2013, February 21–22, 2014 (Fernando Ayala, RL, SK, Santiago Ron), April 8–11, 2018 (Jaime Culebras, Jorge Brito, SK), October 17–30, 2016 (PH, RJM, RL, Paul Maier, Kristiina Ovaska, Amanda Northrup, Bill Langworthy, and one assistant), January 20–30, 2018 (PH, RJM, Amanda Northrup, Nathalie Aall, Bill Langworthy, and two assistants), February 5–13, 2018 (JV, PH, RJM, RL, SK, Jo Bowman, Bill Langworthy, Scott Trageser, and two assistants). Visual encounter surveys were conducted along transects of various lengths within primary forest, secondary and riparian forest, and along streams of various sizes. For smaller streams that had thick vegetation and were too narrow to perform linear transects, we performed general searches of the habitat. During the February 2018 trip (the only survey period when the new species was abundant), surveys consisted of walks along different streams starting at a 1,900 until 2,000 h for nine nights.

Evolutionary relationships. We generated mitochondrial sequences (12S, 16S) for three individuals (ZSFQ 462, 463, 466) of the new species and several other Nymphargus species (Table S1). Extraction, amplification, and sequencing protocols are as described in Guayasamin et al. (2008). The newly obtained sequences (Table S1) were compared with those of all other available species of Nymphargus (see Fig. 2) and all other glassfrog genera, downloaded from GenBank (https://www.ncbi.nlm.nih.gov/genbank/); sequence information and GenBank codes of the outgroups are listed in Guayasamin et al. (2008), Castroviejo-Fisher et al. (2014), and Twomey, Delia & Castroviejo-Fisher (2014). Sequences were aligned using MAFFT v.7 (Multiple Alignment Program for Amino Acid or Nucleotide Sequences: http://mafft.cbrc.jp/alignment/software/), with the Q-INS-i strategy (Katoh & Standley, 2013). MacClade 4.07 (Maddison & Maddison, 2010) was used to visualize the alignment (no modifications were necessary). Maximum likelihood was run in the IQ-TREE 1.5.5 software (Nguyen et al., 2015). The best-fitting nucleotide substitution model was implemented using ModelFinder within IQ-TREE (Kalyaanamoorthy et al., 2017), which groups partitions with the same model and similar rates and simultaneously searches model and tree space; since only mitochondrial sequences were analyzed, they were considered as a single gene (i.e., they evolve as unit—maternal inheritance and no recombination). Node support was assessed via 1,000 ultra-fast bootstrap replicates, a method that shows less bias that other support estimates (Minh, Nguyen & Haeseler, 2013). Ultra-fast bootstrapping also leads to straightforward interpretation of the support values (e.g., support of ≥95% should be interpreted as significant; Minh, Nguyen & Haeseler, 2013).

Figure 2 Inferred mitochondrial phylogeny of the genus Nymphargus, with the positioning of the new species, Nymphargus manduriacu sp. nov.

Clade support values (bootstraps) were obtained as described in Minh, Nguyen & Haeseler (2013). Taxa in blue correspond to sequences added in this study.

Results

Phylogenetic relationships of Nymphargus. Based on the Bayesian Information Criterion, the best-fit model for our dataset was TIM2+F+R5. Rate parameters were estimated as follows: A–C: 5.14658, A–G: 16.73402, A–T: 5.14658, C–G: 1.00000, C–T: 49.89941, G–T: 1.00000. Base frequencies were: A: 0.341, C: 0.255, G: 0.186, T: 0.218. We generated 31 new sequences (Table S1), including species that have never been part of centrolenid phylogenies.

The inferred phylogeny (Fig. 2) confirms the placement of the following species within the genus Nymphargus (sensu Guayasamin et al., 2009): N. balionotus (Duellman, 1981), N. cariticommatus (Wild, 1994), N. lasgralarias Hutter & Guayasamin (2012), N. spilotus (Ruiz-Carranza & Lynch, 1997), and N. sucre Guayasamin (2013). The new species, described below, is also part of the genus Nymphargus. Relationships among Nymphargus species are similar to those reported in previous studies (Guayasamin et al., 2008; Castroviejo-Fisher et al., 2014; Twomey, Delia & Castroviejo-Fisher, 2014), but some novel relationships are revealed because of our increased taxon sampling (Fig. 2).

Species description

Nymphargus manduriacu new species

LSID urn:lsid:zoobank.org:pub:E5C0E7E4-9C69-4830-A514-AD1F4B80311C.

Common names. English: Manduriacu glassfrog. Spanish: Rana de Cristal de Manduriacu.

Holotype. ZSFQ 0466 (Fig. 3), adult male from Reserva Río Manduriacu (0.310755°N, 78.8569°W; 1,215 m), Provincia de Imbabura, República del Ecuador, collected by JV and RJM on February 7th, 2018.

Figure 3 Nymphargus manduriacu sp. nov. in life.

(A)–(C) Adult male, holotype, ZSFQ 0466. (D)–(F) Adult female, paratype, ZSFQ 0462.

Paratypes. ZSFQ 0465, adult male, with same data as holotype. ZSFQ 0462, adult female, and ZSFQ 0463 (Fig. 3), adult male, with same data as holotype, but collected at a different stream (0.310818°N, 78.857°W; 1,230 m) on February 6th, 2018.

Referred material. ZSFQ 0464 (Fig. 4), metamorph, with same data as holotype.

Figure 4 Life stages of Nymphargus manduriacu sp. nov.

(A) Egg clutch (ZSFQ 467). (B) Metamorph (ZSFQ 618).

Generic placement: The new species is placed in the genus Nymphargus Cisneros-Heredia & McDiarmid (2007), as modified by Guayasamin et al. (2009), based on morphological and genetic data. All species in Nymphargus share an absence of webbing among Fingers I–III and absence or reduced webbing between Fingers III and IV; additionally, males lack humeral spines (except N. grandisonae; N. armatus and some populations of N. griffithsi have an enlarged ventral crest on their humeri that can resemble a humeral spine). Nymphargus manduriacu sp. nov. presents all the aforementioned traits and its placement within Nymphargus is unambiguous. Phylogenetic analyses of mitochondrial genes also place N. manduriacu sp. nov. in the genus Nymphargus (Fig. 2).

Diagnosis. Nymphargus manduriacu sp. nov. is distinguished from most glassfrogs by lacking webbing between inner fingers and having, in life, a grayish green dorsum with numerous yellow spots, which sometimes are surrounded by an ill-defined black ring (i.e., false ocelli). On the Pacific slopes of the Ecuadorian and Colombian Andes, there are very few species that share the two aforementioned traits with N. manduriacu sp. nov.; these species are: N. buenaventura, N. ignotus, N. spilotus, and N. luminosus. Differences among these species are summarized in Table 1 and Figs. 5 and 6. The sister species of N. manduriacu sp. nov. is N. balionotus, which is easily differentiated by its unique dorsal color pattern, a green dorsum with several black and occasionally yellow to cinnamon blotches (Duellman, 1981; Arteaga-Navarro, Bustamante & Guayasamin, 2013). Also, the uncorrected p genetic distance between N. manduriacu sp. nov. and N. balionotus is 6.4–6.7% for the 12S and 16S concatenated matrix.

Table 1 Differences between N. manduriacu sp. nov. and similar species from the Pacific Andes of Ecuador and Pacific and Central Andes of Colombia.

	N. manduriacu sp. nov.	N. buenaventura	N. ignotus	N. luminosus	N. spilotus	
Distribution (elevation)	1,215–1,238 m	800–1,200 m	1,280–2,050 m	1,140–1,430 m	1,850–1,940 m	
SVL (adult males)	24.0–25.7 (n = 3)	20.9–22.4 (n = 4)	22.2–25.4 (n = 61)	27.7–30.0 (n = 15)	25.7–26.6 (n = 2)	
Dorsal coloration in life	Grayish green with yellow spots, which are sometimes surrounded by ill-defined rings	Light green with diffuse pale yellow spots	Dorsum pale tan to olive brown with black ocelli surrounding orange or yellow spots	Green with numerous yellow spots (95–217 spots; n = 16)	Olive green with small yellow spots	
Webbing between Fingers III and IV	Basal (Fig. 6A)	Basal	Basal	Basal, but more extended than in other Nymphargus (Fig. 6B)	Basal (Fig. 6D)	
Source	This study	Cisneros-Heredia & Yánez-Muñoz (2007); this study	Rada, Ospina-Sarria & Guayasamin (2017)	Ruiz-Carranza & Lynch (1995); this study	Ruiz-Carranza & Lynch (1997); this study	

Figure 5 Nymphargus manduriacu sp. nov. and similar species.

(A) N. manduriacu, Reserva Río Manduriacu, Ecuador, uncollected. (B) N. buenaventura, Cascadas de Manuel, Cantón El Guabo, Provincia El Oro, 800 m, Ecuador, DHMECN 10982, photo by Juan Carlos Sánchez. (C) N. luminosus, Quebrada la Honda y La Amarill, Verada Venados Arriba, Municipio de Frontino, Departamento de Antioquia, Colombia, MAR 3576, photo by Marco Rada. (D) N. spilotus, Parque Nacional Natural Selva de Florencia, Colombia, JD 060, photo by Jesse Delia.

Figure 6 Hand webbing Nymphargus manduriacu sp. nov. and similar species.

(A) N. manduriacu, ZSFQ 0463, adult male, paratype. (B) N. luminosus, ICN 15930, adult female, holotype. (C) N. spilotus, ICN 35255, adult female, holotype.

Definition. The new species is distinguished from all other Centrolenidae by the following combination of characters: (1) dentigerous process of vomer low or absent, without vomerine teeth; (2) snout truncate in dorsal and truncate to slight rounded in lateral view; (3) tympanic annulus barely evident, lower ¾ visible, tympanic membrane colored as dorsal skin, supratympanic fold present; (4) dorsal skin shagreen, with microspicules in adult males; (5) ventral skin granular, subcloacal area with two large subcloacal warts; (6) parietal peritoneum white, iridophores covering 1/3–1/2 parietal peritoneum (conditions P2 or P3); pericardium white (i.e., covered by iridophores), all other visceral peritonea clear (condition V1); (7) liver lobed and hepatic peritoneum clear (lacking iridophore layer, condition H0); (8) adult males without projecting humeral spine; (9) webbing between fingers I, II, and III absent, basal between fingers III and IV; (10) toe webbing basal between toes I and II, III 1½–(2½–3−) III (1⅓–1½)–(3–3−) IV (3–3−)–(1½–2−) V; (11) lacking dermal ornamentations in the form of tubercles, folds, or fringes on hands, arms, feet, or legs; (12) nuptial excrescences Type I and VI; concealed prepollex; (13) Finger I slightly longer than Finger II; (14) diameter of eye larger than width of disc on Finger III; (15) color in life, grayish green to olive green with yellow spots, which, sometimes, are surrounded an ill-defined black ring (i.e., false ocelli); bones green; (16) color in preservative, lavender dorsum with cream spots; (17) iris coloration in life: light gray with thin gray reticulations and pale yellow hue around pupil; (18) melanophores present and abundant along Fingers III and IV, less dense on Finger II, and rarely present on Finger I, present and abundant along Toes IV and V, less dense on Toe III, only at the base of Toes I and II; (19) males call from upper side of leaves; advertisement call is a high-pitched “chirp,” with a single, pulsed note with a duration of 0.093–0.118 s (x̅ = 0.10 ± 0.007; n = 10) and a dominant frequency at 4,052–4,447 Hz (x̅ = 4,267.7 ± 1,18.3); (20) fighting behavior unknown; (21) egg masses deposited on upper side of leaves, clutch size 15–32 (n = 4); males do not attend or get in contact with clutches; (22) tadpoles undescribed; (23) SVL in adult males 24.0–25.7 mm (n = 3), and in an adult female 28.8 mm.

Description of holotype. Adult male (ZSFQ-0466; Figs. 3–7). Head wider than long (head length 90% of head width); snout truncate in dorsal view and slightly rounded in lateral profile; canthus rostralis indistinct, slightly concave; loreal region concave; lips not flared; nostrils protuberant, closer to tip of snout than to eye, directed frontolaterally; internarial area barely depressed. Eyes large, directed anterolaterally at an angle ~45°; transverse diameter of disc of Finger III 53% eye diameter. Supratympanic fold low, obscuring upper edge of tympanic annulus; tympanic annulus small and almost indistinct, oriented mostly vertically; tympanic membrane colored as surrounding skin. Dentigerous process of vomer absent; choanae rounded; tongue ovoid and unnotched, with ventral posterior fourth not attached to mouth floor; vocal slits extending posterolaterally from about the lateral margin of tongue (at about half the length of tongue) to angle of jaws.

Figure 7 Call of the holotype of Nymphargus manduriacu sp. nov.

(A) Oscillogram. (B) Audiospectrogram. File number: LBE-C-042.

Humeral spine absent. Hand and ulnar folds absent; relative lengths of fingers: III > IV > II > I; webbing absent between Fingers I–III, basal between Fingers III and IV, webbing formula III 23/4–23/4 IV; discs expanded, nearly elliptical; disc pads with triangular shape; subarticular tubercles small, round, simple; supernumerary tubercles numerous, fleshy, giving the palm a warty texture; palmar tubercle elliptical, simple; nuptial pad Type I present but faint, extending from ventrolateral base to dorsal surface of Finger I, covering the proximal half of Finger I.

Length of tibia 59% SVL; tarsal folds absent; two-thirds webbed foot; toe webbing basal between toes I and II, III 1½–3− III 1½–3− IV 3−–1½ V; discs on toes elliptical; disc on Toe IV narrower that disc on Finger III; disc pads triangular; inner metatarsal tubercle large, ovoid; outer metatarsal not evident; subarticular tubercles small, round; supernumerary tubercles absent. Skin on dorsal surfaces of head, body, and lateral surface of head and flanks shagreen with numerous minute spinules; throat smooth; belly and lower flanks granular; cloacal opening directed posteriorly at upper level of thighs; cloacal ornamentation absent except for pair of enlarged subcloacal tubercles.

Color in life. Grayish green to olive green dorsum with yellow spots. Melanophores concentrated around yellow spots, sometimes looking like false ocelli. Upper lip unpigmented. Inner fingers and toes yellowish. Anterior half of ventral parietal peritoneum white, posterior portion translucent. Color of bones green. Iris light gray with thin, dark gray reticulations, and pale yellow hue around pupil (Figs. 3 and 5).

Color in ethanol. Dorsal surfaces gray lavender with small white spots. Parietal peritoneum white, iridophores covering 1/3–1/2 parietal peritoneum. Heart covered by white pericardium; all other visceral peritonea unpigmented.

Variation. The only known female is larger than males and lacks microspicules (Fig. 1). Male holotype was slightly greener than female. Metamorph was uniformly green showing faint light spots. Meristic variation is reported in Table 2.

Table 2 Morphological measurements (in mm) of the type series of N. manduriacu sp. nov.

	ZSFQ-0466	ZSFQ-0463	ZSFQ-0465	ZSFQ-0462	
	Male	Male	Male	Female	
SVL	25.3	25.7	24.0	28.8	
Tibia length	15.0	15.0	14.2	18.0	
Foot length	12.1	11.6	11.0	13.8	
Head length	8.2	8.3	8.1	8.7	
Head width	9.1	8.8	8.7	10.2	
Snout to eye distance	2.8	2.8	3.2	3.7	
Interorbital distance	3.7	3.9	3.9	4.0	
Upper eyelid width	2.2.	2.1	2.2	2.6	
Internarinal distance	2.3	2.4	2.1	2.4	
Eye diameter	3.4	3.2	3.1	3.8	
Tympanum diameter	0.8	0.9	1.0	1.2	
Radioulna length	5.7	5.6	5.5	6.9	
Hand	8.9	9.1	8.5	10.0	
Finger I	4.1	4.4	4.2	5.4	
Finger II	5.5	5.2	4.8	6.1	
Disc of Finger III width	1.8	1.9	1.7	1.8	

Natural history. Although RMR has been visited numerous times during the past years (see Methods), the new species was only regularly detected during February 2018. This was during the wet season, with the site experiencing particularly heavy rains on a daily basis during this time frame. During all previous surveys, only a single individual was observed along a transect in mature secondary forest (0.3144°N, 78.855°W; 1,238 m; October 22, 2016 at 2,036 h). The point where the individual was found was along a narrow, sloping ridge, with the Manduriacu River ca. 20 m east, and a smaller stream (ca. 3–4 m wide) about 15 m west; the individual, uncollected, was perched on a leaf with a perch-height of 350 cm when found.

In February 2018, even though sampling targeted numerous large rivers (ca. 4–7 m wide) and smaller streams (ca. 0.5–2 m wide), all individuals of the new species were found in only two places. The first location is a narrow stream (ca. 0.75 m wide, 0.310818°N, 78.857°W; 1,230 m) with dense vegetation, where on February 6, 2018 at 2,033 h, a male (ZSFQ-0463) was calling and perched on a leaf above the water 280 cm high, and a female (ZSFQ-0462) on a leaf below the male, with a perch height of 170 cm. Conditions were wet, with light rain during the time of capture. The following night, three additional individuals were collected along a nearby, but much larger (ca. 4 m wide) and fast-flowing stream. The first was an adult male (ZSFQ-0465) collected at 1,930 h, found moving on a leaf 220 cm above the ground, and located five m from the stream (i.e., not directly above water). The second individual was a metamorph (ZSFQ-0464; 16.4 mm SVL) collected at 2,000 h, perched on a leaf 100 cm directly above the stream. The third specimen was a male (ZSFQ-0466) collected at 2,200 h, perched on a leaf 400 cm directly above water. This male was 15–30 cm from four egg clutches; these clutches, placed on the upper surfaces of leaves (Fig. 4), contained 26–32 embryos (x̅ = 25.75 ± 7.5883). Finally, an uncollected adult individual was found on February 12, 2018 at 2,311 h along a slightly larger stream (two to three m wide) immediately adjacent to camp on the south side (0.3104°N, 78.858°W; 1,224 m). This uncollected individual was perched in herbaceous vegetation in an area where a large treefall was lying across the stream; the individual was found 0.5 m from the stream, with a perch high of 65 cm.

Call (Fig. 7). The following description is based on the recording of two males (ZSFQ 0465-0466) obtained by JV on February 05, 2018 at the type locality. Each call is a high-pitched “chirp” that consists of a single note and has a duration of 0.093–0.118 s (x̅ = 0.10 ± 0.007; n = 10). Notes are pulsed (8–12 pulses per note; x̅ = 10.33 ± 1.366). In each call, there is a very slight increase in the dominant frequency with time; the dominant frequency is at 4,052–4,447 Hz (x̅ = 4,267.7 ± 118.3). Time between calls is 3.9–8.6 s (x̅ = 5.72 ± 1.82). Among closely related species to the new taxon (Fig. 2), only the call of N. grandisonae is described (Hutter et al., 2013). The call of N. manduriacu sp. nov. is differentiated mainly by having a higher dominant frequency (4,052–4,447 Hz in N. manduriacu sp. nov., 3,100–4,048 Hz in N. grandisonae).

Evolutionary relationships of the new species. Given the gene and taxon sampling of our study, Nymphargus manduriacu sp. nov. is sister to N. balionotus. The latter taxon was considered to be as incertae sedis within the subfamily Centroleninae (Guayasamin et al., 2009). Here, we formally place Centrolenella balionota Duellman, 1981 in the genus Nymphargus sensu Guayasamin et al., 2009. Nymphargus manduriacu sp. nov. and N. balionotus are endemic to the Pacific slopes of the northern Andes, and are found syntopically at RMR.

Distribution. Nymphargus manduriacu is only known from a few nearby streams within the Río Manduriacu Reserve (0.31°N, 78.85°W), Imbabura province, on the Pacific slopes of the Andes of Ecuador (Fig. 1). Based on these limited records, the species occupies a narrow elevational range of 1,215–1,242 m.

Conservation status. We recommend that N. manduriacu should be considered as Critically Endangered, following IUCN (2001) criteria B2a (known to exist from a single locality) and B2biii (continuing decline, observed, inferred or projected, in area, extent and/or quality of habitat). The main threats for the species are habitat destruction and contamination associated with cattle ranching, agriculture and, most seriously, mining activities (see Discussion; Fig. 8). Although RMR is still poorly surveyed, northwestern Ecuador has been the target of intense herpetological research (e.g., Lynch & Duellman, 1973, 1997; Arteaga-Navarro, Bustamante & Guayasamin, 2013; Arteaga et al., 2016), including areas nearby RMR (i.e., Reserva Los Cedros; Hutter & Guayasamin, 2015); thus, the restricted distribution of the new species is, most likely, real.

Figure 8 Map of Río Manduriacu Reserve (Imbabura province, Ecuador), with conservation plans and mining threats.

Etymology. The specific epithet “manduriacu” is a noun in apposition and refers to the type locality of the species, Río Manduriacu Reserve, a conservation area managed by Fundación EcoMinga (https://ecomingafoundation.wordpress.com/).

Discussion

Biogeographic patterns within Nymphargus. The phylogeny we present (Fig. 2) reveals a number of biogeographic patterns that are worth highlighting. Given the current taxon sampling (60% of the described diversity of the genus), all sister species within Nymphargus are geographic neighbors, including some sister taxa that, in some areas, are sympatric (e.g., N. manduriacu/N. balionotus, N. griffithsi/N. lasgralarias). On the Amazonian slope of the Andes, most sister species have allopatric distributions: N. aff. chancas/N. mariae, N. cariticommatus/N. sucre, N. cochranae/N. sp, N. siren/N. sp, N. pluvialis/N. posadae. The only two sampled Nymphargus species distributed on the eastern slope of the Cordillera Central of Colombia (N. spilotus/N. rosadus) are also sister species. These biogeographic patterns are congruent with the hypothesis that recent speciation in Nymphargus is mediated by the linearity of the Andes, which results in elongate geographical ranges and reduces potential contact and gene flow among parapatric populations (Remsen, 1984; Graves, 1988). It is also possible that dry river-valleys play a role as barriers (see Krabbe, 2008; Guayasamin et al., 2010; Arteaga et al., 2016; Prieto-Torres, Cuervo & Bonaccorso, 2018; Winger & Bates, 2015). Examples of dry Andean river valleys in Ecuador include the Mira, Guayllabamba, Jubones, Girón, and Paute. To date, however, there are no explicit studies designed to test the effect of the mentioned valleys on diversification processes.

Biodiversity value of Río Manduriacu Reserve. In Ecuador, RMR is a very atypical site for vertebrate diversity. It is the only known place that houses an extant reproducing population of the long lost and presumed-extinct Tandayapa Andean Toad (Rhaebo olallai; Lynch et al., 2014), and the Mindo Cochran frog (N. balionotus; R. Maynard, 2016–2018, personal observation). Prior to their rediscovery at RMR, the former had not been seen in over four decades and the latter in over a decade in Ecuador. Additionally, there is a large number of other threatened amphibians, including the critically endangered Centrolene ballux and the endangered Pristimantis crenunguis, Pristimantis pteridophilus, and Pristimantis scolodiscus, as well as several undescribed anurans (e.g., Noblella sp, Pristimantis spp). The reserve also serves as a stronghold for other Critically Endangered animals, such as the brown-headed Spider Monkey (Ateles fusciceps), included on the list of the 25 most endangered primates in the world (Mittermeier et al., 2007; Schwitzer et al., 2017). Other mammals recorded at RMR include the Spectacled Bear (Tremarctos ornatus), Pacarana (Dinomys branickii), and Oncilla (Leopardus tigrinus), all classified as Vulnerable (IUCN VU) (S. Kohn, 2015–2018, personal observation). Also notable is the presence of jaguars (Panthera onca; S. Kohn, 2016, personal observation), as their populations along the western slopes of the Andes are considered Critically Endangered, despite having a global status of Near Threatened (Espinoza et al., 2011).

Threats to the Río Manduriacu reserve

Mining. Immediate threats to the forests at RMR make the conservation of this newly described species, and the biodiversity in the area, a difficult challenge. The primary threat comes from mining concessions (Fig. 8) given by the government to Cerro Quebrado, a subsidiary to the Australian BHP Hillinton, the world’s largest mining company. Ecuador’s legislation requires that any mining operation must consult with local communities and landowners prior to any mining activity (Article 398, Constitución del Ecuador, 2008). Despite the fact that Cerro Quebrado did not consult the local population or local landowners, the Ecuadorian government granted the company a concession to extract gold and copper through an open pit mine (see mining cadaster: http://geo.controlminero.gob.ec:1026/geo_visor/). Thus, with this documentation in hand, it is clear that the mining concession in RMR and nearby areas are void and that mining activities should be prohibited until Cerro Quebrado and the Ecuadorian government abide by the Ecuadorian constitution.

Logging. Illegal and uncontrolled logging also pose a grave threat. Local communities have relied historically on logging as one of their main sources of income. Nonetheless, this logging is mostly illegal (e.g., without government permits or using a permit for one area to extract wood from a different area) and remains poorly regulated. As a result, forest cover in the area surrounding RMR has been reduced drastically in the last two decades (http://mapainteractivo.ambiente.gob.ec/portal/). This problem is exacerbated by the generalized lack of legal land ownership titles.

Conservation actions. Numerous local landowners have protected the forest of RMR for several decades. In 2010 several tracts of forest were incorporated into the Socio Bosque program, a conservation initiative by the Ecuadorian government (http://sociobosque.ambiente.gob.ec/). Through Socio Bosque, landowners and communities that are willing to conserve their forests get financial incentives if they maintain the original forest cover. Starting in 2015 Fundación EcoMinga signed an agreement with the owners of several lots to control, protect and manage the reserve. Through this agreement all funds from Socio Bosque are directed to fund salaries of park rangers and reserve managers. IUCN Netherlands has supported EcoMinga in purchasing new plots in order to expand the reserve. With this new purchase the land belonging to RMR has grown to almost 600 ha, while the plots affiliated to Socio Bosque, but outside the limits of the reserve, cover an additional 350 ha. EcoMinga’s medium and long term goal is to purchase nearby land that is not under any conservation program.

In early 2017, EcoMinga began a community project to promote sustainable alternatives (e.g., ecotourism, revenues from academic projects, sustainable agriculture, vanilla farming) to wood extraction (training locals as guides, mainly for birdwatching tourism). The aim is to generate a model for community work that is appropriate for the country and that can later be replicated in other reserves. This program has faced difficulties since recently local community members were hired by Cerro Quebrado to illegally enter into RMR. This has created constant tension within the communities and with stakeholders.

Conclusions

We provide morphological, genetic, and acoustic evidence that support the validity of a new species, N. manduriacu. Also, we infer a new mitochondrial phylogeny of the genus Nymphargus that allows us to reveal speciation patterns in this taxon, mainly that recent speciation events in this genus seem to be heavily influenced by the linearity of the Andes and dry river-valleys that are run transversal to this mountain range. Finally, the new species is considered as Critically Endangered because of its restricted distribution, habitat destruction and contamination associated with cattle ranching, agriculture and, most seriously, mining activities. At Río Manduriacu Reserve, mining has become one of the most dangerous threats to biodiversity, especially to species with highly restricted distributions.

Appendix I: Examined specimens

Nymphargus buenaventura: Ecuador: Provincia de El Oro: Cantón El Guabo: Cascadas de Manuel, 800 m, DHMECN 10982; Cantón Piñas: Reserva Buenaventura (03°38′S, 79°45′W, 1,200 m), DHMECN 3563 (holotype), 2524, 3561–62 (paratypes).

Nymphargus balionotus: Ecuador: Provincia de Imbabura: Río Manduriacu Reserve (0.31°N, 78.85°W; 1,215–1,238 m), ZSFQ 0531–533.

Nymphargus griffithsi: Ecuador: Provincia de Pichincha: Río Saloya, 1219 meters, BMNH 1940.2.20.4 (holotype), BMNH 1940.2.20.3 (paratype); Reserva Las Gralarias, MZUTI 100, 102, and 099, “Hercules Giant Tree Frog Creek,” (0°01.529′S, 78°42.243′W; 2,175 m); MZUTI 101, “Five Frog Creek,” (0°01.870′S, 78°42.358′W; 2,150 m); MZUTI 098, “Heloderma Creek” (0°01.245′S, 78°42.370′W; 2,200 m).

Nymphargus ignotus: Colombia: Departamento del Valle del Cauca: Municipio de La Cumbre, Corregimiento de Bitaco, Vereda Chicoral, tributary of Río Bitaco (03°34′09.9″N, 76°35′42.7″W, 1,950 m), ICN 55799–800, ICN 21524–5; Peñas Blancas, Farallones de Cali, ca. six km by road SW of Pichindé (04°53′05.2″N, 76°08′52.5″W, 1,900 m), ICN 14748, holotype, ICN 14749–77; Municipio de Dagua, Finca San Pedro, headwater of the Quebrada La Seca, eight km S of Queremal (03°28′30,1″N, 76°42′10.8″W, 1,940–2,050 m), ICN 41333–41. Departamento de Chocó: Municipio de San José del Palmar, 12–12.6 km on the San José del Palmar-Cartago road (03°24′59.8″N, 76°37′12.6″W, 1,850 m), ICN 19641. Departamento de Risaralda: Municipio de Mistrató, km 10–11 carretera Mistrató-San Antonio del Chami, quebrada Mampay (05°21′N, 75°52′W, 1,760 m), ICN 30040–8, ICN 30056.

Nymphargus lasgralarias: Ecuador: Provincia de Pichincha: Reserva Las Gralarias, MZUTI 096 (holotype), MZUTI 091–095, 097 (paratypes).

Nymphargus luminosus: Colombia: Departamento de Antioquia: Municipio de Frontino: Corregimiento Nutibara: Km 23–27 on the Nutibara-La Blanquita road, 1,000–1,430 m, ICN 15930 (holotype), ICN 15918–20, 15922–29, 15931–33 (paratypes).

Nymphargus spilotus: Colombia: Departamento de Caldas: Municipio de Samaná, Corregimiento Florencia: sitio “Rancho Quemado,” 1,940 m, ICN 35155 (holotype); sitio El Estadero, 1,850 m, ICN 35157–58 (paratypes); zona “El Estadero” (o “Rancho Quemado”), ICN 38073 (paratype).

Supplemental Information

Supplemental Information 1 Mitochondrial sequences generated in this study, with associated information (species name, museum number, locality and Genbank number).

Click here for additional data file.

The article benefited from reviews by two anonymous reviewers, and comments from Elisa Bonaccorso and Mario H. Yánez-Muñoz. The Biodiversity Group is grateful to Nathalie Aall, Jo Bowman, Bill Langworthy, Paul Maier, José María Loaiza, Amanda Northrup, Kristiina Ovaska, and Scott Trageser for assistance in the field. For access to collection specimens, we thank the Natural History Museum and Biodiversity Research Center of University of Kansas, Instituto de Ciencias Naturales of the Universidad Nacional de Colombia, and the Museo de Zoología of the Universidad Tecnológica Indoamérica. Sebastián Montilla and Carolina Reyes-Puig provided valuable assistance when we examined specimens at ICN and the ZSFQ, respectively. Sequences and photos of N. spilotus (JD 060) and Nymphargus sp. (LSB 210) were graciously provided by Marco Rada and Jesse Delia. We also thank Juan Carlos Sánchez for providing the photo of N. buenaventura.

Additional Information and Declarations

Competing Interests

Author Contributions

Animal Ethics

Field Study Permissions

DNA Deposition

Data Availability

New Species Registration

José Vieira is employed by Tropical Herping, Sebastián Kohn is employed by the non-profit organization Fundación Cóndor Andino, Ryan L. Lynch is employed by the non-profit organization The Third Millennium Alliance, Paul S. Hamilton and Ross J. Maynard are employed by the non-profit organization The Biodiversity Group, and Sebastián Kohn is employed by the non-profit organization Fundación EcoMinga.

Juan M. Guayasamin conceived and designed the experiments, performed the experiments, analyzed the data, contributed reagents/materials/analysis tools, prepared figures and/or tables, authored or reviewed drafts of the paper, approved the final draft.

Diego F. Cisneros-Heredia conceived and designed the experiments, performed the experiments, contributed reagents/materials/analysis tools, authored or reviewed drafts of the paper, approved the final draft.

José Vieira performed the experiments, prepared figures and/or tables, authored or reviewed drafts of the paper, approved the final draft.

Sebastián Kohn conceived and designed the experiments, performed the experiments, contributed reagents/materials/analysis tools, prepared figures and/or tables, authored or reviewed drafts of the paper, approved the final draft.

Gabriela Gavilanes performed the experiments, authored or reviewed drafts of the paper, approved the final draft.

Ryan L. Lynch performed the experiments, contributed reagents/materials/analysis tools, authored or reviewed drafts of the paper, approved the final draft.

Paul S. Hamilton conceived and designed the experiments, performed the experiments, contributed reagents/materials/analysis tools, authored or reviewed drafts of the paper, approved the final draft.

Ross J. Maynard conceived and designed the experiments, performed the experiments, contributed reagents/materials/analysis tools, authored or reviewed drafts of the paper, approved the final draft.

The following information was supplied relating to ethical approvals (i.e., approving body and any reference numbers):

Research was conducted under permits NoMAE-DNB-CM-2015-2017 and 018-2017-IC-FAU-DNB/MAE, issued by the Ministerio del Ambiente del Ecuador. The study was carried out in accordance with the guidelines for use of live amphibians and reptiles in field research (Beaupre et al., 2004), compiled by the American Society of Ichthyologists and Herpetologists (ASIH), the Herpetologists’ League (HL) and the Society for the Study of Amphibians and Reptiles (SSAR).

The following information was supplied relating to field study approvals (i.e., approving body and any reference numbers):

Field research was authorized by the Ministerio del Ambiente de Ecuador.

The following information was supplied regarding the deposition of DNA sequences:

All sequences generated in this study are deposited in GenBank (Table S1).

The following information was supplied regarding data availability:

Morphological measurements are available in Table 2.

The following information was supplied regarding the registration of a newly described species:

Publication LSID: urn:lsid:zoobank.org:pub:E5C0E7E4-9C69-4830-A514-AD1F4B80311C

Nymphargus manduriacu LSID: urn:lsid:zoobank.org:act:57564C1C-4636-4DB4-ADDC-7BB23DFBE86D.

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
