# Peer review of "A new glassfrog (Centrolenidae) from the Chocó-Andean Río Manduriacu Reserve, Ecuador, endangered by mining"

_PeerJ, doi:10.7717/peerj.6400_

## Round 0.1 · original submission · Minor Revisions

Dear Dr. Guayasamin and colleagues:

Thanks for submitting your manuscript to PeerJ. I have now received two independent reviews of your work, and as you will see, both are very favorable. Well done! Nonetheless, both reviewers raised some relatively minor concerns about the research, and areas where the manuscript can be improved.

Two things in particular: 1) please address the issues raised regarding speciation concepts and all aspects of phylogeny estimation, and 2) please provide a more thorough Introduction in your revised manuscript (much more background and information is required).

I agree with the issues raised by the reviewers, and thus feel that their concerns should be adequately addressed before moving forward.

Therefore, I am recommending that you revise your manuscript accordingly, taking into account all of the issues raised by the reviewers. I do believe that your manuscript will be ready for publication once these issues are addressed.

Good luck with your revision,

-joe

Reviewer 1 ·

Basic reporting

No comment.

Experimental design

No comment.

Validity of the findings

This is a well-written and concise description of a new species of Nymphargus from Ecuador. I am not entirely convinced that the new species they describe is valid and I think more work is needed to explain how the new species differs from closely related species. I think that the new species is probably new (judging from branch lengths on the tree), but some key details are missing to convince me. I think that addressing these two issues would greatly improve the ms:

1) The comparisons with other species seem to be under-developed and constrained to mostly color characters. It is unclear what species concept is being used here, and what evidence presented indicates the species is new. Genetic distances are not mentioned, which would be important to get a sense for how closely related the new species is to its sister species. Importantly, there is no comparison with N. balionotus in the ms, so it is difficult to know how the new species differs from its evolutionary sister species, which occurs syntopically. This seems like a key comparison and should be made directly in the “Comparison with similar species” section.
2) The discussion of speciation patterns in Nymphargus in the Andes is too strongly stated given no biogeographic or other analyses, and should be framed as speculation. The conclusions here are drawn from observations on the phylogeny, with support coming from sister species being close geographically. However, there are many other Nymphargus that are not mentioned that could be contrary to these conclusions, and I think a future study would be necessary to figure out the drivers of speciation in this group. Additionally, many of the sister species pairs mentioned in the discussion occur syntopically, so it seems like the situation is fairly complex.

Additional comments

Minor corrections/suggestions:

Line 74: “patters” = “patterns”
Line 116: Could the specimens examined be moved to an appendix?
Line 160: can delete “between them”

Results:

Line 200: How many partitions did you use?
Line 247: The table does not show very clearly how the species all differ. Are there other characters that can be used to compare? It also seems like call characters could be used in this section as well. If there are no calls available for the other species, than that should be clarified.

Discussion:

The numerous headings and subheading here is confusing, with different formats for each (i.e. a new line for some, colon for others). Authors should make the heading format consistent between sections.


Figures and Tables:

Figure 1: the white mainland around Ecuador looks awkward. I would suggest either removing the blue ocean or giving the white mainland a grey or other coloration.

Figure 2: the blue tip coloration explanation was cut off in the figure caption?

Reviewer 2 ·

Basic reporting

This is a well-done species description of a new glassfrog species from essentially a single locality in Ecuador. The authors went above and beyond in that they included sequence data from a number of Nymphargus species that had not been sequenced previously. The phylogeny is notable in that (1) it clearly supports the new species, (2) it resolves the status of "Cochranella" balionota as a member of Nymphargus, and (3) suggests an interesting biogeographic pattern of parapatric sister species, which the authors discuss.

The English is clear and unambiguous, references are apt, and the article structure is appropriate. The figures in particular are quite nice.

My major comment (aside from some minor stuff I've included as comments in the word file) is that the introduction is too short and doesn't do a great job at setting up the study. The intro has 6 sentences total, and one of the sentences is discussing species that are not even the focus of the paper. I would make the following suggestions: (1) elaborate paragraph 1 to include more background and history on Nymphargus, in particular N. balionota. (2) Elaborate on paragraph 3 so that rather than describing results from the study, the focus is shifted more towards biogeographic hypotheses (i.e. elaborate on dry river valleys in particular, give some possible examples, etc).

Experimental design

The paper meets all the relevant standards.

Validity of the findings

The validity of the new species is not in doubt and is supported by morphological and phylogenetic evidence. Aside from that the paper provides a substantial contribution to glassfrog systematics.

Additional comments

Most of my comments are given on the word document, which I'll be emailing to the editor.

Annotated reviews are not available for download in order to protect the identity of reviewers who chose to remain anonymous.

---

## Round 0.2 · accepted · Accept

Dear Dr. Guayasamin and colleagues:

Thanks for re-submitting your manuscript to PeerJ, and for addressing the concerns raised by the reviewers. I now believe that your manuscript is suitable for publication. Congratulations! I look forward to seeing this work in print, and I anticipate it being an important resource for the communities studying frogs and conservation biology. Thanks again for choosing PeerJ to publish such important work.

-joe

#